# CFD Numerical Simulation in Building Drainage Stacks as an Infection Pathway of COVID-19

**DOI:** 10.3390/ijerph19127475

**Published:** 2022-06-18

**Authors:** Cheng-Li Cheng, Yen-Yu Lin

**Affiliations:** Department of Architecture, National Taiwan University of Science and Technology, Taipei 10607, Taiwan; ccl@mail.ntust.edu.tw

**Keywords:** building drainage system, fluid simulation, contaminant controlling, infection risk, indoor contamination, pandemics, water trap

## Abstract

Being aware of global pandemics, this research focused on the potential infection routes in building drainage systems. Case studies have found that dysfunctional building drainage systems not only failed to block contaminants but also potentially became a route for the spreading of viruses. Several fluid simulations in pipelines were conducted in this research using COMSOL Multiphysics. In particular, virus transmission from one patient’s room to other uninfected residential units through pipelines was visualized. A 12-story building, which is commonly seen in the local area, was designed as a simulation model to visualize the transmission and analyze its hazards. Furthermore, five environmental factors were filtered out for discussion: distance, time span, pressure, initial concentration, and environment temperature. By manipulating these factors, the relationship between the factors and the behavior of the contaminant could be explored. In addition, a simulation with a different pipeline arrangement was included to observe the virus diffusion behavior under different scenarios. The visualized simulation concluded that the contaminant would spread through the drainage system and arrive at the neighboring four floors within an hour under the circumstances of a 12-story building with broken seals and constant pressure and contaminant supply on the seventh floor. Meanwhile, the whole building would be exposed to infection risks by the continuous virus spreading through a drainage system. Distance, time span, and pressure were considered critical factors that affected indoor contamination in the system. On the other hand, initial concentration and environmental temperature did not have significant roles. Visualizing the behavior of viruses provides a glimpse of what happens behind walls, paving the way for recognizing the importance of maintaining functional drainage systems for individuals’ health.

## 1. Introduction

Building drainage systems are well recognized as one of the essential facilities in building service engineering. However, many substantial issues regarding health and indoor contamination in many existing buildings should be highlighted. The gravity drainage system is commonly used worldwide, and the appliance trap seal plays a vital role in safeguarding occupied spaces from stench and vermin spreading from the drainage network [1]. Furthermore, the inappropriate design of the drainage system within existing buildings can result in sanitary problems, including air transient caused by discharges in the drainage stack and trap seal depletion [2]. People have become more aware of the importance of drainage system design and maintenance due to the SARS epidemic in 2003 [3]. The rapid spread of the SARS virus at the Amoy Gardens housing complex in Hong Kong provided a great lesson regarding the potential health risks attributed to drainage systems by highlighting the cross-contamination route caused by appliance trap seal depletion [4].

A couple of years have passed since the COVID-19 outbreak, taking countless lives and reshaping human behaviors. As a response, studies have been conducted in different scopes and research fields to prevent further virus dispersion. From the architectural perspective, this research focused on visualizing virus diffusion in building drainage systems while treating dysfunctional water traps through simulation using Computational Fluid Dynamics. In addition, five environmental factors were examined concerning their influence on the system, and an additional simulation with different pipeline arrangements is also included in this article. Furthermore, several fluid simulations for pipelines were conducted in this research using COMSOL Multiphysics. In particular, the virus transmission from one patient’s room to other uninfected residential units through pipelines was visualized. This research mainly visualized the movement within the pipelines through computer simulation and observed how the contaminant diffused through the drainage system. On the basis of related literature, five environmental factors—distance, spreading time, pressure, contaminant concentration, and room temperature—were examined in this article concerning their influences on the system. Moreover, the simulation results in this study demonstrate that contamination may occur when water traps are left impaired.

Conventional construction often hides building drainage systems behind walls. Hence, the quality of drainage systems is easily neglected. Rarely maintained pipelines could lead to water trap abnormalities, leakage, drying, or blockage. More issues can also arise due to dysfunctional drainage systems. For instance, during the SARS outbreak, the Amoy Gardens in Hong Kong experienced a massive infection resulting from its dysfunctional drainage system. With its incomplete water traps, virus stacking inside the drainage system reached other rooms without warning. Consequently, the virus infected over 300 people without face-to-face interaction.

Drainage systems refer to the pipelines transporting domestic water within the building. The main driving power source of drainage systems in most countries is gravity. Moreover, water traps are common devices consisting of a single stack of water that remains within the pipes; their main purpose is to act as an obstacle between the clean and contaminated water. A breaking water seal is a situation in which air from two sides of the trap connects, resulting in indoor contamination. Such an incident may be caused by poor drainage system design, lack of maintenance, sudden pressure shifts [5,6,7,8], or long-term misuse. In order to reduce the effect of pressure shifting, a special pipe joint and a double stack pipe system (Figure 1) are applied to drainage systems in residential buildings.

The Amoy Garden incident occurred as a result of a lack of maintenance and long-term misuse. When mistreated seals are broken, viruses diffuse through the drainage system and spread from one room to another. Furthermore, exhaust systems, which are an increasingly common installation in modern society, accelerate the process by pumping the contaminated air out of the pipes [9,10]. Therefore, this research mainly aimed to visualize and examine the mechanisms of virus dispersion to prevent further contamination incidents.

## 2. Materials and Methods

### 2.1. Computational Fluid Dynamics (CFD)

CFD is a scientific research method achieved through the computational ability of computers. By dividing fluid into a grid, the computation of each point provides information about the movement of the fluid [11,12,13,14,15,16]. Since the outbreak of COVID-19 and SARS, governments and academics have placed research emphasis on viruses [17,18,19,20,21,22,23]. The CFD tool provides information on the possible paths of virus dispersion in confined areas [24]. However, only a few existing studies have used CFD to analyze the relationship between virus dispersion and building drainage systems. This study used COMSOL as a simulation tool to determine plausible virus spreading routes in building drainage systems. First, the research examined references and cases to ensure the risk of contaminant spreading when dealing with broken water seals. Second, the formula used by the simulation software was studied, and the environmental factors determined by this study were filtered out. Third, a simulation model and environment were constructed to represent actual scenarios. Furthermore, the magnitude of the effects of these factors on the system was determined by manipulating them [25,26,27]. The mechanism of contaminant diffusion was also visualized through simulation. Ultimately, each environmental factor’s influence and its potential risks in the neighboring rooms were concluded [28,29].

### 2.2. Mathematical Simulation Formula

This study was a combination of fluid behavior and concentration distribution. The simulation software calculated the information mathematically by employing physics formulas and controlled boundary conditions. For the fluid behavior, the Navier–Stokes Equation and the Conservation of Mass were used as the base simulation principles.

**Theorem** **1.**
*Navier–Stokes Equation.*



(1)
$$\rho \frac{\partial u}{\partial t}+\rho \left(u\cdot \nabla \right)u=-\nabla P+\rho g+\mu \nabla^{2}u$$


**Theorem** **2.**
*Conservation of Mass.*



(2)
ρ∇ · (u)=0


The Convection Diffusion Equation and Mass Transfer Equation were introduced to the system for the concentration distribution.

**Theorem** **3.** 
*Convection Diffusion Equation.*



(3)
$$\frac{\partial c}{dt}+\nabla \;\cdot \;\left(-D\nabla c\right)+u\;\cdot \;\nabla c=R$$


**Theorem** **4.**
*Mass Transfer Equation.*



(4)
N=−D∇c+uc


### 2.3. Numerical Solution Strategy

In this study, COMSOL Multiphysics was the software used to construct the simulation. The CFD software is verified as a trustworthy simulation tool because of the notable amount of research based on it [30,31]. In the simulation, a standard ‘k-ε’ model was employed in the system as a turbulence model. The discretization scheme used to solve the governing equation was the finite element method (FEM). Moreover, the iterative method was introduced to the system as the numerical method to solve non-linear algebraic equations. The tolerance was controlled by physics with a scaling ratio of 0.05. With the help of mesh generation, the continuous flow domain could be broken down into a discrete grid of cells. A tetrahedron was the basic element organized to fit in the system. Figure 2 shows a display of when a mesh grid was applied to the geometry. The mesh convergence test served as a validation of the simulation, in which we rearranged the system with three different grid densities—normal (2,553,002 elements), coarse (848,980 elements), and coarser (255,440 elements)—and observed the concentration ratio on the sixth floor’s center. The results are shown in Figure 3, which shows that the mesh density did not have a great impact on the system.

### 2.4. Model Construction and Environmental Condition

When it comes to building drainage systems, a vast number of methods with different pipeline designs are being used depending on local conditions. One system worth mentioning is the double stack system, a drainage system design in which a ventilation pipe is connected to a soiled-water pipe or a wastewater pipe, which is widely used internationally because of its ability to balance pressure differences. In addition, various variations of the double stack system were designed to fulfill different needs of all kinds of buildings. Thus, it is not practical to simulate every scenario in a single study. In this research, we focused on a typical fitting with a domestic drainage system in residential buildings.

A 12-story building community, commonly seen locally, with two households on each floor was designed. Each household had one bathroom with dimensions of 3 m × 2 m × 3 m, including two floor openings: a wastewater pipe and a soiled pipe, as shown in Figure 4. Households were divided into two groups, A and B; the 7th floor of side A was set up as the contaminant source, and the two groups were connected through the main horizontal pipe below the 1st floor, as shown in Figure 5a. Under the worst scenario, a situation was assumed in which the water trap dried out thoroughly, so the pipelines were completely connected. In the simulation, carbon dioxide from the contaminant source was set as the carrier of the virus [32]. The mechanism of the spread of the virus could be determined by observing the behavior of carbon dioxide. Furthermore, the initial concentration was set at 1000 ppm, and the experiment assumed a constant contaminant supply so the source could maintain its concentration and to ensure that there would be no other reaction if the contaminant was consumed. Moreover, a pushing force of 0.05 Pa was assumed to act as the pressure shift in the system. The span of the simulation was set for 3 days.

Additionally, another simulation was conducted under an assumed circumstance in which the two sides shared one set of vent pipes (see Figure 5b), which would provide information on the horizontal movement of the contaminant and the risk of horizontal units.

### 2.5. Environmental Factors

Environmental factors were considered independent variables. A series of inputs was set up. Then, the adjustment of the system was observed, and the influence of the factors on the system was studied. In this study, five factors were included and examined through simulation and discussion.

(1)Distance

The distance between the source and the target unit was considered crucial. Therefore, this paper focuses on the differences between each floor. The floor is introduced as a distance unit in the discussion.

(2)Time span (t)

The time span was also considered a critical factor. A three-day simulation was conducted, and the changes in the system on an hourly basis were observed.

(3)Pressure (P_0_)

Pressure was introduced to the system, considering the popularity of the exhaust system and the constant shifting of flushing. The simulation focused on the exhaust system maintaining a constant pressure supply. Its effects on the system could be determined by adjusting the pressure force.

(4)Initial concentration (C_0_)

The initial concentration was the CO_2_ concentration applied to the source unit. Theorem 3 shows that a higher concentration difference would reinforce a faster diffusion rate.

(5)Temperature (T)

Temperature was expected to influence fluid density. Thus, the environmental temperature was adjusted, and then the reaction of the model system was observed.

## 3. Results

### Simulation and Analysis

The simulation results revealed how the virus diffused into each unit and how far it influenced the system. In order to make a comparison among the different circumstances, the contaminants’ concentration distribution is expressed in concentration ratio (C ratio), that is, the local concentration divided by the initial concentration (C_0_) on the seventh floor. In addition, simulation result screenshots were collected at the end of every day. The results are shown in Figure 6. Furthermore, it was shown that the contaminant reached each unit on side A within a day. The concentration in each cell was also raised gradually over time until reaching its terminal concentration, which was similar to the initial concentration. It was also observed that, as a consequence of setting the seventh floor as the middle line, units tended to have similar concentrations to those with the same distance between the middle line. Moreover, the results also revealed that the further the concentration was from the middle line, the lower the concentration.

Figure 7 shows the snapshot of the first three hours, indicating the diffusion speed. In particular, the contaminant had already reached the unit three floors away within an hour.

After adjusting the pressure applied to the source, a handful of data was obtained concerning the pressure’s influence on the system. The results on the third day under different pressures are shown in Figure 8. It was found that pressure had a large influence on the system. A decreasing P_0_ caused the virus to spread more slowly and narrowly, while an increasing P_0_ worsened the contamination of the whole system.

Furthermore, the change in the initial concentration was addressed. However, contrary to the study’s assumptions, the adjustment of C_0_ did not have a significant direct difference on the system, as shown in Figure 9. It also revealed that the initial concentration was not an important influence factor.

Figure 10 indicates how environmental temperature influenced the system under three different atmospheric temperatures in different seasons. It shows that different environmental temperatures did not result in significant system reactions.

Furthermore, the contamination mainly occupied side A and had less effect on side B. In order to examine the risk of being contaminated by horizontal units, a scenario was introduced in which both sides shared one vent pipe. The results are shown in Figure 11. The fluid in the pipes was not restrained from moving vertically. Hence, the virus traveled to the units within its reach.

## 4. Discussion

The simulation results showed how the contaminant spread in such a short period. The virus reached the units three floors away from the source within an hour. In addition, it reached six floors within three days and raised the concentration in each unit simultaneously. With the visualized simulation as evidence, we can confirm the high risk of broken water seals, resulting in contaminant diffusion.

Environmental factors may influence the system in different ways. By manipulating the parameters, the impact of each factor could be studied. First, distance was critical to virus transportation. The farther away a unit was, the farther the virus had to travel to reach it; hence, the contamination was slower and less severe. Second, time was found to be a crucial parameter in the system. Once the spreading began, the neighboring units started to become infected; thus, the longer the time span, the higher the concentration was within units until it reached its terminal concentration, which was as high as the initial concentration. Third, pressure played an important role as the driving force in the spread of the virus. A higher pressure reinforced faster virus spreading. The central concentration data in each unit was collected from side A on the third day and analyzed, as shown in Figure 12. This presents how the pressure accelerated the transportation process and raised the average concentration of the system. Employing the Navier–Stokes Equation, pressure and fluid velocity were found to have a positive correlation.

Fourth, the results showed that the factor of the initial concentration did not perform as expected. There were no radical changes when the number was adjusted. This phenomenon can be explained through the Convection Diffusion Equation, as we assumed no reaction in the system. Therefore, the driving force landed on the diffusion from the concentration difference or the transportation from pressure. According to the data, transportation overpowered diffusion; the latter could not make a significant difference. Therefore, it was concluded that the system could be driven by pressure.

Fifth, the environmental temperature did not significantly affect the system. Despite adjusting the atmospheric temperature, the system did not indicate different fluid behaviors.

Therefore, an alternative pipeline was designed in which both sides of the building shared the same set of venting pipes. The results showed that the contaminated target gas could reach side B quickly without going through the main horizontal pipe. It was observed that the virus spread to the side B units in the first hour of the simulation, indicating the possibility of horizontal virus spread.

## 5. Conclusions

This article sought an alternative method in which a virtual drainage system was reconstructed through computational simulation to inspect the risk of a particular scenario. The simulation results not only indicate the high possibility of potential risks but also show how a virus can spread in such a short time. Under certain circumstances, the contaminant could spread three floors away within an hour. In addition, pipelines made it possible for the virus to reach every connected unit in the building. Environmental factors, including time, distance, and pressure, played a crucial role in influencing the system. On the other hand, the initial contaminant concentration has less of an impact than other concentrations. Notably, pressure tended to be the main driving force in the spread of the virus. Additionally, a broken water trap posed a risk of a virus breach in the residential area, which could cause community infection. Visualizing the behavior of viruses provides a glimpse of what happens behind walls, paving the way for recognizing the importance of maintaining functional drainage systems for individuals’ health.

## Figures and Tables

**Figure 1 ijerph-19-07475-f001:**
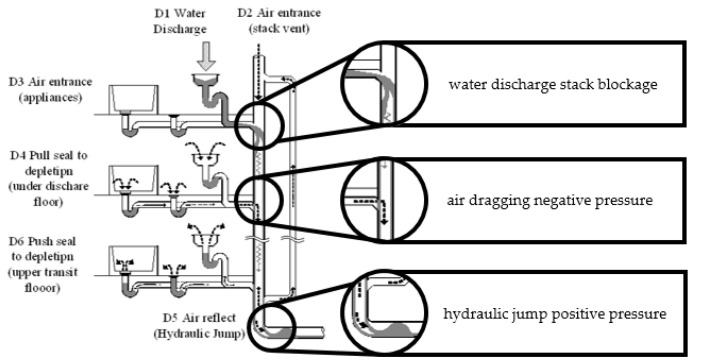
Mechanism of double stack pipe drainage system design in residential buildings.

**Figure 2 ijerph-19-07475-f002:**
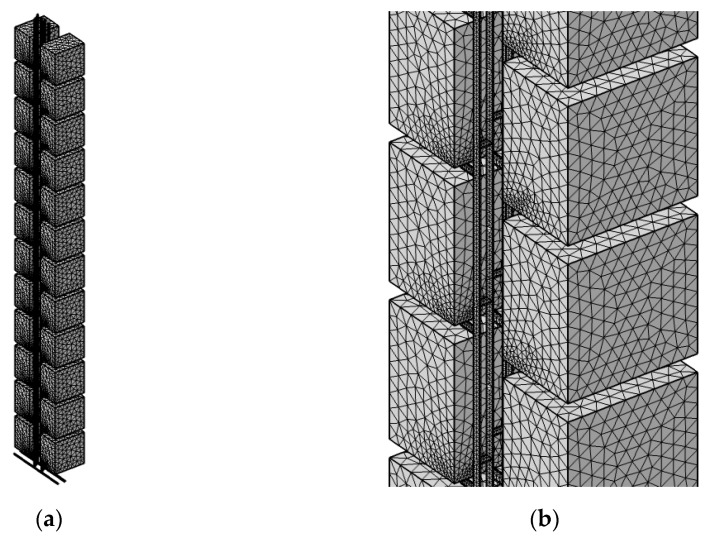
(**a**) System mesh grid generation. (**b**) Mesh generation over units.

**Figure 3 ijerph-19-07475-f003:**
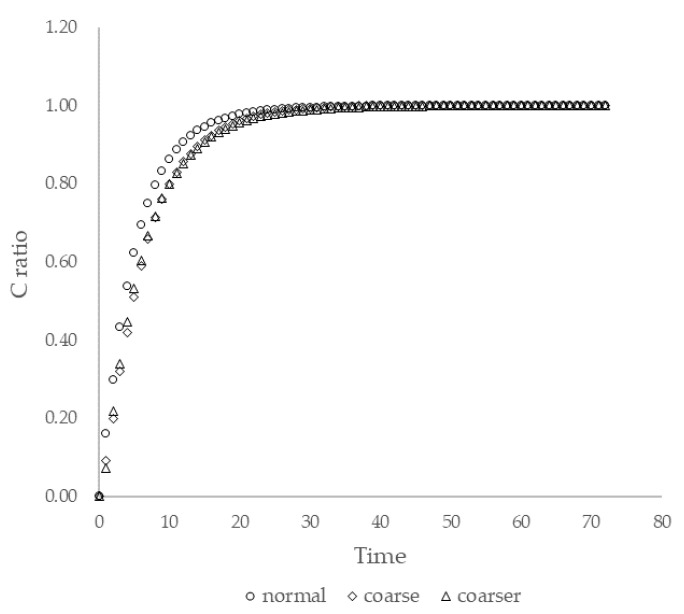
Mesh test using C ratio as benchmark.

**Figure 4 ijerph-19-07475-f004:**
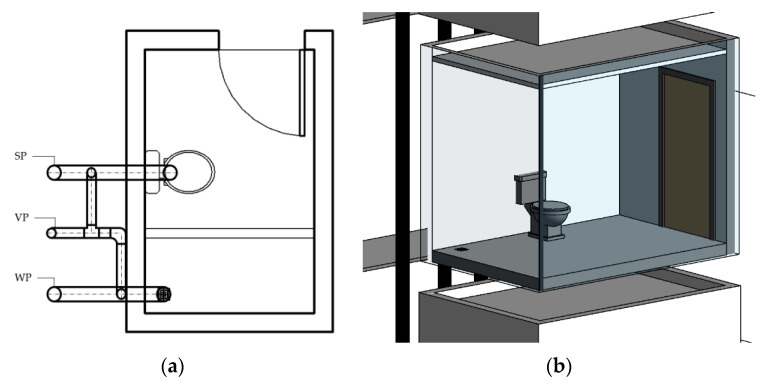
Simulation model unit design including unit opening and pipelines. (**a**) Simulation unit floor plan. (**b**) Bathroom unit perspective drawing.

**Figure 5 ijerph-19-07475-f005:**
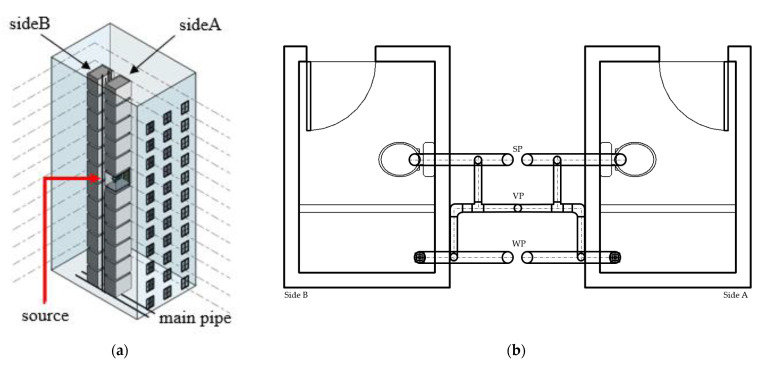
Simulation model design. (**a**) System overlook. (**b**) Scenario of two bathroom units.

**Figure 6 ijerph-19-07475-f006:**
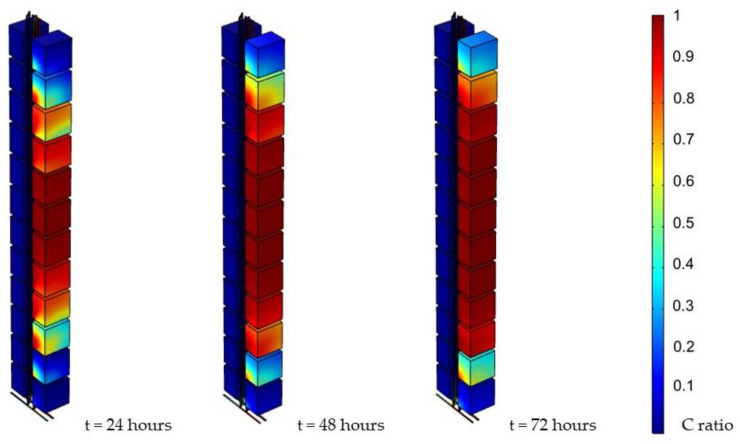
Simulation results (0.05 Pa, 1000 ppm).

**Figure 7 ijerph-19-07475-f007:**
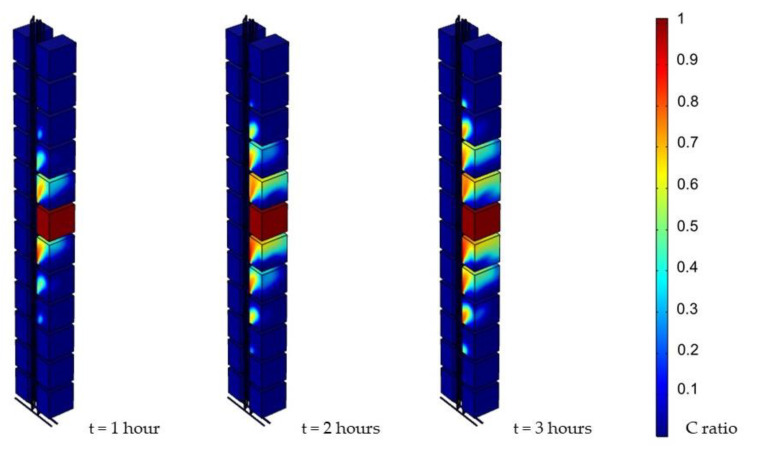
Simulation results of first three hours (0.05 Pa, 1000 ppm).

**Figure 8 ijerph-19-07475-f008:**
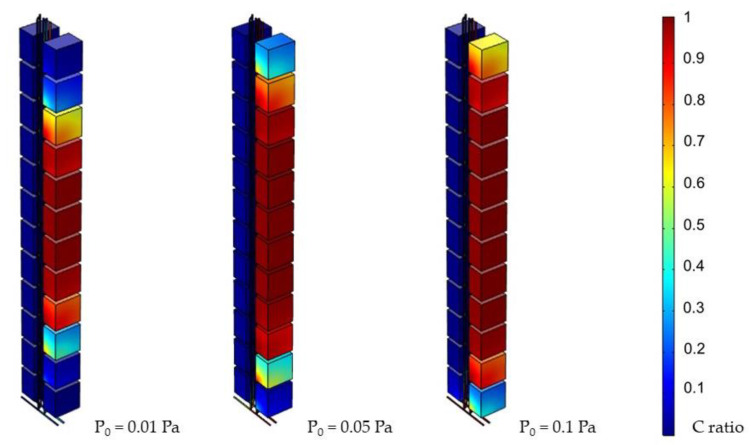
Simulation results for various P_0_ (day 3, 1000 ppm).

**Figure 9 ijerph-19-07475-f009:**
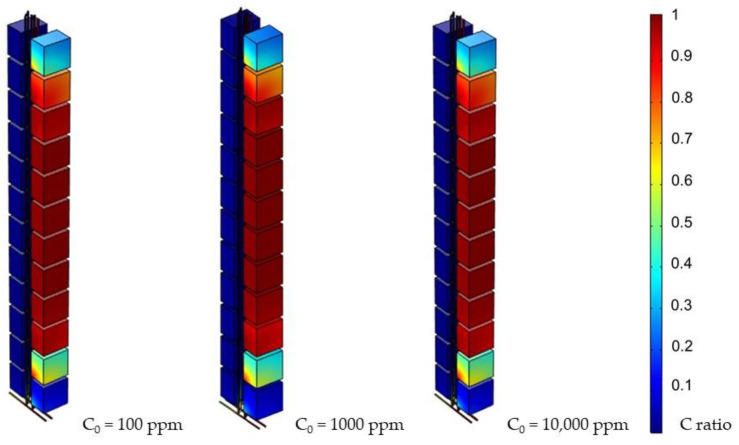
Simulation results for various C_0_ (day 3, 0.05 Pa).

**Figure 10 ijerph-19-07475-f010:**
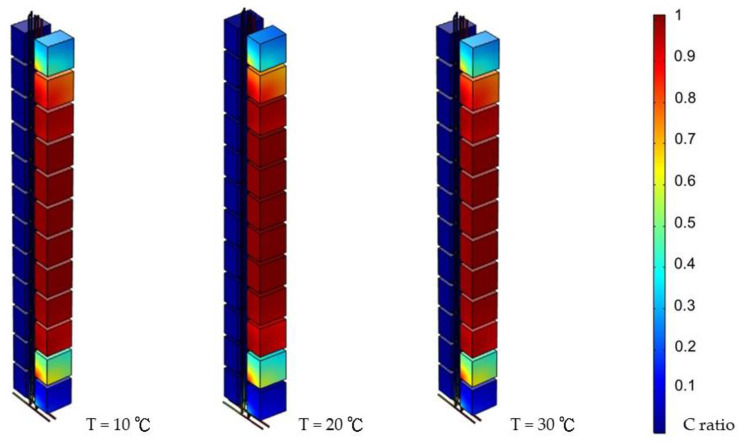
Simulation results for various T (day 3, 0.05 Pa).

**Figure 11 ijerph-19-07475-f011:**
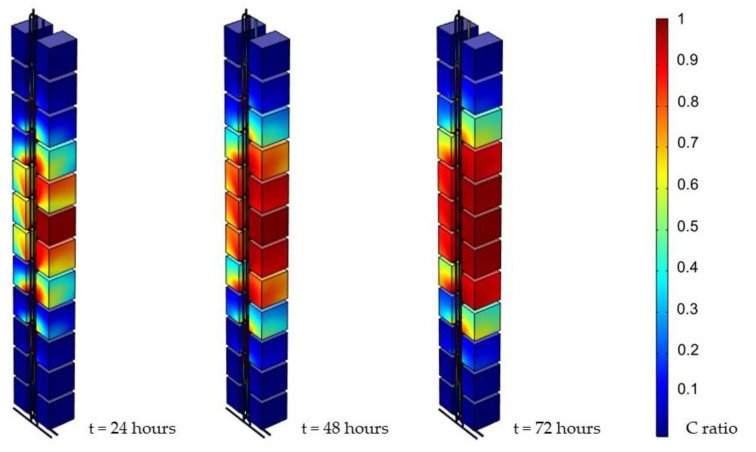
Simulation results of alternative scenario (0.05 Pa, 1000 ppm).

**Figure 12 ijerph-19-07475-f012:**
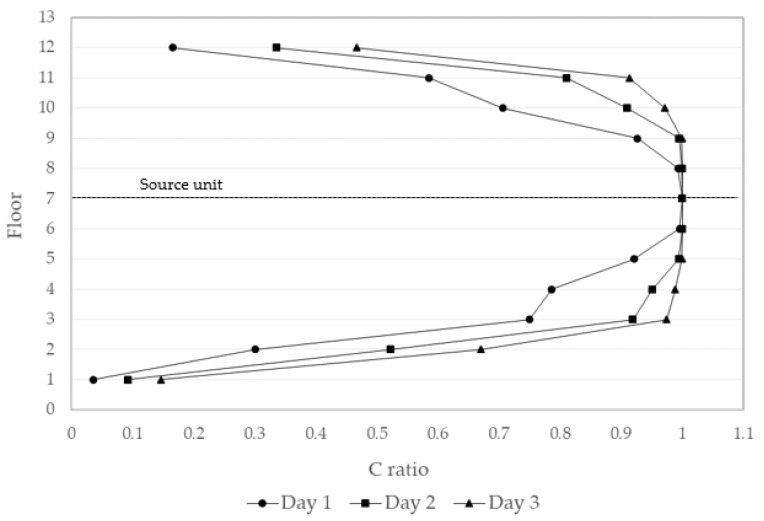
Central concentration in units with different pressures (day 3, 1000 ppm).

## Data Availability

Not applicable.

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
