# Peer review of "CFD Numerical Simulation in Building Drainage Stacks as an Infection Pathway of COVID-19"

_ijerph, 2022, doi:10.3390/ijerph19127475_

Round 1
Reviewer 1 Report
A brief summary
The research presented showed a potential spreading of contaminants through the drainage system. This topic is very actual and can predict the possible way of spreading not only COVID 19 but another contaminant. Several computational fluid dynamics simulations were solved to visualise the spreading in pipelines. Results showed that contaminants would spread through the drainage system and arrive at the neighbouring four floors within an hour under certain assumptions.
General concept comments
Results from the research described the very idealized situation when all water traps are dysfunctional. Statistically, this situation is unlikely, only if it is a targeted act. As authors referred to the Amoy Gardens in Hong Kong. I think it was the interplay of several circumstances. The results should be validated with experimental or in Situ measurements, to achieve real spreading.
Specific comments
A constant contaminant supply is questionable.
Authors could give the information about turbulence model, time step, and mesh grid.
There is a mistake at line 183. Not Figure 7, but Figure 4.
The quality of the figures was poor and should be improved.
Author Response
Please see the attachment. Authors appreciate much for the suggestions and understanding.

Reviewer 2 Report
Reviwer comments for ijerph-1740045
"CFD Numerical Simulation in Building Drainage Stacks as an Infection Pathway of COVID-19". 13/5/22
Abstract, Line 23: "certain assumptions" - list the parameters or factors.
Page 2, Line 58: "five environmental factors" - list the parameters or factors.
Page 2, bottom of page and Figure 1: this Figure should be clearer - it shows both single stack and double stack systems. The hydraulic jump is valid, but more common is simple turbulence caused by the bend radius. Either could present the temporary blockage shown. The partial stack blockage caused by higher discharges (i.e. S1 and D1) could also cause pressure transients.
Page 4, bottom of page, Figure 3: are the blue pipe interconnections staggered and/or swept? This should be explained. A very unusual arrangement.
Page 4, Line 162: if CO2 means carbon dioxide, it should be written CO2 (subscript 2).
Page 9, Line 227: remove 'We' - "We abstracted the central concentration data in each unit....." becomes "The central concentration data in each unit was abstracted.....".
General comments.
A potentially useful paper which establishes a technique. The drainage ventilation arrangements are specified, but this should be explained in context by comparing to other examples. In particulat, the 'X' shaped vent interconnects on page 4 are unusual. While it is not expected that the authors model every scenario, other ventilation arrangements used internationally should be explained, and it should be highlighted that the model adopted is just one example. It should also be explained that updrafts caused by 'stack effect' (higher wind velocity with height and temperature effects) often create an updraft when the system is quiet: therefor the contamination pathways could be amplified, especially with unusual 'X' pattern interconnections. With these changes, the value of this aubmission will be increased.
Author Response

(The authors gave the same response as above.)

Reviewer 3 Report
The authors present an interesting study which is particularly relevant given the ongoing pandemic. The article has many grammatical errors which need to addressed. Following are a few in the abstract and introduction. There are many more throughout the article.
1) line 10: "With the advent..." change this line. Advent is not the right word to use here.
2) line 15: "A commonly seen 12-story...": mention commonly seen in the "local" area?
3) line 19: "mathematically discussed" change to "could be explored"?
4) line 23: "Meanwhile, the whole building ....": Not clear what this line means. Continuous diffusion meaning with diffusion continuing for longer periods?
5) line 47: "Studies have been..." check grammar.
6) line 50: "based on Computational Fluid ..." change "based" to "using"
7) line 58: "under a restrained situation". reword this.
8) There are many more....
Comments:
1) This paper is missing a validation section. The authors need to demonstrate that the simulations they are running represent reality.
2) The paper is also missing a numerical methods section which should come after section 2.2
3) Figure 6: Are these pressure values realistic?
4) I encourage the authors to take a look at other papers that use CFD to get an idea on how to write these missing sections.
Author Response

(The authors gave the same response as above.)

Round 2
Reviewer 3 Report
This version of the manuscript has seen some improvement. My primary concern is that the study still lacks a validation section.
"The CFD software is verified as a trust-worthy simulation tool because of the noticeable amount of research based on it [31][32]. " Is not enough to ensure that the results obtained are correct. This is why you see CFD papers will have a grid convergence study/boundary layer comparisons/turbulence model comparisons/reason for turbulence model selection/skin friction comparisons/pressure gradient comparisons. All this may not apply to the current problem but some of them do. Most commercial CFD solvers (if not all) can give very different results based on the settings used. I request the authors follow: https://www.grc.nasa.gov/www/wind/valid/tutorial/tutorial.html
Verification/Validation needs to be done no matter the simulation package and at the start of every new CFD problem.
In its current form the results from this paper are at best qualitative. Quantitative comparisons can only be made once the validity of the results has been established.
Author Response
Please see the attachment. Authors appreciate for the reminder and understanding
